# Structural Similarity Loss for Learning to Fuse Multi-Focus Images

**DOI:** 10.3390/s20226647

**Published:** 2020-11-20

**Authors:** Xiang Yan, Syed Zulqarnain Gilani, Hanlin Qin, Ajmal Mian

**Affiliations:** 1School of Physics and Optoelectronic Engineering, Xidian University, Xi’an 710071, China; hlqin@mail.xidian.edu.cn; 2School of Science, Edith Cowan University, Joondalup, WA 6027, Australia; s.gilani@ecu.edu.au; 3Computer Science and Software Engineering, The University of Western Australia, Crawley, WA 6009, Australia; ajmal.main@uwa.edu.au

**Keywords:** multi-focus image fusion, convolution neural network, unsupervised learning, structural similarity

## Abstract

Convolutional neural networks have recently been used for multi-focus image fusion. However, some existing methods have resorted to adding Gaussian blur to focused images, to simulate defocus, thereby generating data (with ground-truth) for supervised learning. Moreover, they classify pixels as ‘focused’ or ‘defocused’, and use the classified results to construct the fusion weight maps. This then necessitates a series of post-processing steps. In this paper, we present an end-to-end learning approach for directly predicting the fully focused output image from multi-focus input image pairs. The suggested approach uses a CNN architecture trained to perform fusion, without the need for ground truth fused images. The CNN exploits the image structural similarity (SSIM) to calculate the loss, a metric that is widely accepted for fused image quality evaluation. What is more, we also use the standard deviation of a local window of the image to automatically estimate the importance of the source images in the final fused image when designing the loss function. Our network can accept images of variable sizes and hence, we are able to utilize *real* benchmark datasets, instead of simulated ones, to train our network. The model is a feed-forward, fully convolutional neural network that can process images of variable sizes during test time. Extensive evaluation on benchmark datasets show that our method outperforms, or is comparable with, existing state-of-the-art techniques on both objective and subjective benchmarks.

## 1. Introduction

Most imaging systems, for instance, digital single-lens reflex cameras, have a limited depth-of-field: such that the focus is on the scene content that is in the near vicinity of the imaging plane. Specifically, objects closer to or further away from the focal point appear to be blurred or unfocused in the image. Multi-Focus Image Fusion (MFIF) aims at reconstructing an “in-focus” image from two or more partly or “out-of-focus” images of the same scene. For example, there is a student standing in front of the library building, we take a photo of this student with a common camera, we will obtain one image with a clear student and a blurry library building and the other image with a clear library building and a blurry student by different focus positions. To obtain an image with a clear student and library building (all-in-focus), the multi-focus image techniques can do it. MFIF techniques have wide ranging applications in the fields of surveillance, medical imaging, computer vision, remote sensing and digital imaging [1,2,3,4,5].

The advent of Convolutional Neural Networks (CNNs) has seen a revolution in computer vision: in tasks ranging from object recognition [6,7], semantic segmentation [8,9], action recognition [10,11], optical flow [12,13] to image super-resolution [14,15,16]. Recently, Prabhakar et al. [17] used deep learning to fuse multi-exposure image pairs. This was followed by Liu et al. [18], who proposed a Convolutional Neural Network (CNN), as part of their algorithm, to fuse multi-focus image pairs. That algorithm learns a classifier to distinguish between the “focused” and “unfocused” images to calculate a fusion weight map. Later, Tang et al. [19] improved the algorithm by proposing a pixel-CNN (p-CNN) for categorization of “focused” and “defocused” pixels in a pair of multi-focus images. Yang et al. [20] proposed a multi-level features CNN architecture to yield the fused image.

It is known that the performance of CNNs is often a function of large amounts of labeled training data [21]. Liu et al. [18] and Tang et al. [19] addressed this problem by simulating blurred versions of benchmark datasets (datasets used for image recognition). Unfocused images were generated by adding Gaussian blur in randomly selected patches, making their training dataset unrealistic. Furthermore, since their method is based on calculating weight fusion maps, after learning a classifier, it does not provide an end-to-end solution. Hence, the algorithm requires some post-processing steps for improving the results. Another issue is that, in most well known deep networks (e.g., DeepFuse, Deepface and Facenet [17,22,23]), the input image size is restricted to the training image size. For instance, DeepFuse [17] creates fusion maps during training and requires the input image size to match the fusion map dimensions. This problem is circumvented by sliding a window over the image and obtaining patches to match the fusion map size. These patches are then averaged to obtain the final weight fusion map of the same size as corresponding source images, thereby introducing redundancy and errors in the final reconstruction.

To address these issues, we present an end-to-end deep network trained on benchmark multi-focus images. The proposed network takes a pair of multi-focus images and outputs the all-focus image. We train our network in an unsupervised fashion, without the need for ground truth ‘all focused’ images. This method of training requires a robust loss function. In contrast to the existing unsupervised multi-focus image fusion approaches, our loss function computes only the Structural Similarity (SSIM) of input images, which is widely used to measure the similarity between the fusion image and the focused image, but not add other loss functions such as adversarial and gradient difference. We use the standard deviation of a local window of the image to automatically estimate the importance of the source images in the final fused image when determining the formulation of our loss function, since it can be used to measure the sharpness of input images. We conduct an extensive experimental validation with a comparison to the state-of-the-art methods using widely used datasets. Experimental results verify the effectiveness of the proposed method, and it also can compare to or is better than the state-of-the-art methods.

## 2. Related Work

The literature is rich in research on multi-focus image fusion. The bulk of the research work can be divided into two domains: namely, ‘transform domain’ and ‘spatial domain’ based algorithms [24]. The spatial domain based algorithms have become popular owing to the advent of CNNs. Here, we present a brief overview of the two types of image fusion techniques, followed by a short overview of advancements in deep learning in this field:

**Transform domain based multi-focus image fusion.** Image fusion has been extensively studied in the past few years. Earlier methods were mostly based on the transform domain, owing to their intuitive appeal. This research mainly focuses on ‘pyramid decomposition’ [25,26], ‘wavelet transform’ [27,28] and ‘multi-scale geometric analysis’ [1,29]. Common methods used for multi-focus image fusion include the ‘gradient pyramid’ [26], ‘discrete wavelet transform’ [30], ‘non-subsampled contourlet transform’ [29], ‘shearlet transform’ [31] and ‘curvelet transform’ [32]. In all of these approaches, the source image is first decomposed into a specific multi-scale domain (represented by some coefficients), then all the corresponding decomposed coefficients are integrated to generate a series of comprehensive coefficients. Finally, these coefficients are reconstructed by performing the corresponding inverse multi-scale transform. In the transform domain based multi-focus image fusion, the selection of the multi-scale transform approach is significant, but at the same time, the fusion rules for high and low-frequency coefficients also cannot be ignored, since they directly affect the fusion results.

In the recent past, under the broad category of transform based methods, ‘Independent Component Analysis (ICA)’, ‘Principal Component Analysis (PCA)’, ‘Higher-Order Singular-Value Decomposition (HOSVD)’ and sparse representation based methods have also been introduced in the field of multi-focus image fusion. The core idea of these methods is to search for a feature space that can efficiently represent the activity of ‘un-focused’ image parts. Hence, the measurement of how ‘focused’ the output is, plays a crucial role in these techniques.

**Spatial domain based multi-focus image fusion.** Algorithms of this category have received significant attention because they do not require the source multi-focus images to be converted to alternate forms. These algorithms exploit different fusion rules to generate an “all-in-focus image”, and can be divided into two categories: ‘pixel based’ and ‘block (or region) based’ algorithms [24]. The latter type of algorithms has been more commonly adopted despite the fact that they suffer from blocking effects in the final fused image. Furthermore, the ‘measure of focus’ is vital to both fusion methods. The pixel based methods for multi-focus image fusion preserve the spatial consistency of the resulting image and also extract vital information from the source ‘defocused’ image pairs. Consequently, they have started attracting increased attention from the research community [33]. Methods incorporating ‘image matting’ [34], ‘guided filtering’ [35], ‘dense scale-invariant feature transform’ [33], ‘edge model and multi-matting’ [36] and ‘content adaptive blurring’ [37] are common examples of ‘pixel based’ spatial domain multi-focus image fusion methods. These methods have achieved competitive results with high computational efficiency.

**Deep learning for multi-focus image fusion.** Deep learning has enabled researchers to learn the ‘focus measure’ from the data instead of hand crafting it. This measure is learned from each patch of the source image, which is fed separately to the neural network. The quality of being ‘data-driven’ produces fewer artifacts in the resulting ‘all-focused’ image and hence makes CNN based methods more robust as compared to their conventional counterparts. Lately, Liu et al. [18] proposed a deep network as a subset of their multi-focus image fusion algorithm. They sourced their training data from popular image classification databases, and simply added Gaussian blur to random patches in the image to simulate multi-focus images. Their CNN classifies focused and unfocused pixels, and generates an initial focus map from this information. The final all-focus image is generated after some post-processing of the initial focus map. This step increases the computational cost, and makes this method less suitable for single GPU processing.

Following Liu et al. [18], Tang et al. [19] proposedta p-CNN fortmulti-focus image fusion, while a multi-level features CNN (MLFCNN) was proposed by Yang et al. [20]. Both methods leverage Cifar-10 [38] to generate training image sets for their networks. They simulate ’defocus’ by adding blur to the original focused images. The model outputs three probabilities: namely the probabilities of defocused, focused or indeterminate for each pixel. These probabilities are used to determine the fusion weight map. These methods also need crucial post processing steps to obtain a desired fusion weight map.

Most recently, with the U-net being successfully applied in image-to image translation [39] and pixel-wise regression [40]; a U-net based end-to-end multifocus image algorithm was introduced in [41]. This method also needs the ground truth for training the U-net fusion network model. To do this, several unsupervised multi-focus image fusion networks were proposed in [42,43,44].

In contrast to the methods discussed above, our proposed deep network is trained ‘end-to-end’ and does not rely on post processing. Our network is trained on real multi-focus image pairs and utilizes a *‘no-reference quality metric’* called ‘multi-focus fusion structural similarity (SSIM)’, as a loss function (to achieve end-to-end unsupervised learning). What is more, compared with the unsupervised multi-focus image fusion methods, our proposed loss function used the standard deviation of a local window of the image to automatically estimate the importance of the source images in the final fused image, and our method obtained encouraging results. Our model has three components: feature extraction, feature fusion and reconstruction. The proposed method is described in detail in the succeeding paragraphs.

## 3. Deep Unsupervised Network for MFIF

Our goal is to reconstruct a ‘fused image’ that is all-in-focus. Given an input multi-focus image pair, our model produces an image with all pixels in focus. As is well-known, the learning ability of CNN relies heavily on both the network architecture and the loss function. An elementary CNN architecture consists of a succession of convolutional layers connected in a sequential manner. We propose a deep unsupervised Multi-Focus image Fusion Network (*MFNet*). Our *MFNet* design is motivated by the observation that a multi-focus image pair has different features at ‘high’ and ‘detailed’ levels. Thus, to obtain a fused image with all pixels in focus, we leverage the features at various levels and propose a novel fusion strategy. As a first step we fuse the high level features of the input image pair by averaging them and feed the result to a convolutional layer. Information from low level features is leveraged by passing the input image pair through deeper convolutional networks, separately. The features from the low level networks as well as the high level networks are fused and passed through a reconstruction sub-network to obtain the desired fused image. In the following sub-sections, we describe the design of our proposed network (*MFNet*).

### 3.1. Network Architecture

We propose a deep unsupervised model for the generation of focused images through multi-focus image fusion. The network architecture is illustrated in Figure 1 and comprises four main sub-networks: three feature extraction sub-networks and one feature reconstruction sub-network. Our model takes an unfocused image pair as the input and predicts the image with ‘all focused’ pixels.

#### 3.1.1. Feature Extraction Sub-Network

As displayed in Figure 1, each input image from the multi-focus image pair is passed through a feature extraction network (shown in purple) to obtain high-dimensional non-linear feature maps. However, before passing through this network, the images are convolved with a standard 3×3 kernel and 64 output channels. (The literature survey [45] has shown that a kernel of size 3×3 is a good choice for extracting low level features in the initial layers of a deep learning network). The output of the feature extraction network is passed through another convolutional layer without an activation function. Absence of an activation function avoids non-linearity and helps in retaining more semantic information from the input image. The features from these networks for the two images are then fused (by averaging them) to obtain a feature map. We also take the average of the two multi-focus image pairs and pass this image through a different feature extraction network (shown in orange in Figure 1). This step enriches the features by adding more structural information from the input pair. The output of this network is then added to the fused output from the first two feature extraction networks and passed to the feature reconstruction sub-network.

The details of the feature extraction sub-networks are given in Figure 2. Each network is composed of multiple convolutional and rectification layers sans any pooling layer. We use different architectures for the feature extraction sub-networks to cater for the level of features we want to extract, for example, we use deeper networks for high level feature extraction and shallow networks for low level features. The network that takes in the average of the multi-focus images as input has D2 layers while the networks that extract features from individual images of the input pair have D1 layers. We have color coded the networks in Figure 1 and Figure 2 for ease of cross referencing.

#### 3.1.2. Feature Reconstruction Sub-Network

The goal of this module is to produce the final fused image. It takes as input the output of the third feature extraction sub-network and the convolutional features obtained from the two added input images. As shown in Figure 2, the feature reconstruction network also consists of a cascade of CNNs and is deeper than the feature extraction sub-networks. It comprises seven layers, out of which the first six include the ‘Leaky Rectified Linear Units (LReLUs)’ while the last one is followed by a sigmoid non-linear activation function. Using LReLU in the last layer reduces the contrast of the reconstructed image; hence, we replace it with a sigmoid function. The output fused image is produced by the final convolutional layer.

### 3.2. Loss Function

Our unsupervised training strategy does not require ground truth and instead uses the image Structure SIMilarity (SSIM) [46] quality metric. The SSIM is a widely used perceptual image quality metric, which is well aligned with how the Human Visual System evaluates the quality of images. Moreover, the standard deviation (SD) of the pixel intensities is also a classical multi-focus image fusion evaluation metric. Thus, in order to obtain a visually pleasing fusion result, it is natural to use these two metrics to design the loss function. Let the input image pair be denoted by $x_{1}$, $x_{2}$, and let θ represent the network parameters we wish to optimize. The goal of training is to learn a mapping function *g*, which produces an image (fused image) $\hat{y}=g\left(x_{1},x_{2};\theta \right)$ that resembles the desired image (all the pixels in this image are in focus) *y* as close as possible. We now briefly explain the concept of multi-focus SSIM and then present our proposed loss function.

The image structure similarity (SSIM) is designed to calculate the structural similarities of different sliding windows in their corresponding positions between two images. Let *x* and *y* be the reference image and a test image, respectively, and then the SSIM can be defined as: (1)$$\mathrm{SSIM}\left(x,\hat{y}|\omega \right)=\frac{\left(2{\bar{\omega }}_{x}{\bar{\omega }}_{y}+C_{1}\right)\left(2\sigma_{\omega_{x}\omega_{y}}+C_{2}\right)}{\left({\bar{\omega }}_{x}^{2}+{\bar{\omega }}_{y}^{2}+C_{1}\right)\left(\sigma_{\omega_{x}}^{2}+\sigma_{\omega_{y}}^{2}+C_{2}\right)},$$
where $C_{1}$ and $C_{2}$ are two small constants, $\omega_{x}$ is a sliding window in *x*, ${\bar{\omega }}_{x}$ represents the average of $\omega_{x}$, $\sigma_{\omega_{x}}^{2}$ is the variance of $\omega_{x}$ and $\sigma_{\omega_{x}\omega_{y}}$ denotes the covariance of $\omega_{x}$ and $\omega_{y}$. The variables $\omega_{y}$, ${\bar{\omega }}_{y}$ and $\sigma_{\omega_{y}}$ have the same meanings for image *y* (instead of *x*). The value of $\mathrm{SSIM}\left(x,\hat{y}|\omega \right)\in \left[-1,1\right]$ measures the *‘similarity’* between $\omega_{x}$ and $\omega_{y}$. When its value is 1, it means that $\omega_{x}$ and $\omega_{y}$ are the same.

To assess the image quality in the local windows, we first calculate the structure similarities $\mathrm{SSIM}\left(x_{1},\hat{y}|\omega \right)$ and $\mathrm{SSIM}\left(x_{2},\hat{y}|\omega \right)$ using Equation (Equation 1). The constants $C_{1}$ and $C_{2}$ are set as $1\times 10^{-4}$ and $9\times 10^{-4}$ (as in [46]), respectively. The size of the sliding window is 7×7, and it moves by one pixel as it slides from the top-left to the bottom-right of the image. We use the structural similarity of the input images as the matching metric. When the standard deviation std$\left(x_{1}|\omega \right)$ of a local window of input $x_{1}$ is equal to or larger than the corresponding std$\left(x_{2}|\omega \right)$ of input $x_{2}$, it means that the local window image patch of input $x_{1}$ is likely more focused. Therefore, we define an objective function to evaluate the image patch similarity, as follows:(2)$$S\left(x_{1},x_{2},\hat{y}|\omega \right)=\left\{\begin{array}{c} \mathrm{SSIM}\left(x_{1},\hat{y}|\omega \right),\\ if\;\mathrm{std}\left(x_{1}|\omega \right)\geq \mathrm{std}\left(x_{2}|\omega \right)\\ \mathrm{SSIM}\left(x_{2},\hat{y}|\omega \right),\;\mathrm{otherwise} \end{array}\right.$$

Based on the value of $S\left(x_{1},x_{2},\hat{y}|\omega \right)$ in local window ω, we obtain a robust loss function to optimize the unsupervised network. The overall loss function is defined as
(3)$$Loss\left(x_{1},x_{2},\hat{y}\right)=1-\frac{1}{N}\sum_{\omega =1}^{N}S\left(x_{1},x_{2},\hat{y}|\omega \right),$$
where, total number of sliding windows are indicated by *N*. The network is trained by back-propagating the computed loss. Equation (Equation 3) shows that SSIM encodes the structural similarity between the input and the output fused images, and hence performs better than other objective functions.

### 3.3. Implementation Details

All the convolutional layers have 64 filters of size 3×3 in our proposed *MFNet*. We randomly initialized the parameters of convolutional filters and used zero padding match filter sizes. We used LReLUs [47] with a negative slopetof 0.2 as the non-lineartactivation function except for the last convolutional (reconstruction) layer where we chose sigmoid as the activation function. For the feature extraction and reconstruction sub-networks, the number of convolutional layers D1, D2 and D3 are set as 5, 6 and 7, respectively.

We used 60 pairs of multi-focus images from the benchmark Lytro Multi-focus Image dataset [48] and grayscale image pairs from a popular multi-focus image dataset [33,49] as our training data. Since the dataset is small, we randomly cropped 64×64 patches to form our final training dataset. The total number of the cropped patches is 50,000. The ratio of the size of the training, validation and test datasets is 3:1:1. We used Tensorflow [50] to train our model and ran it for 90 epochs. In addition, we set the weight decay to $10^{-4}$, initialized the learning rate to $10^{-3}$ for all layers, set the decay coefficient to $10^{3}$, the decay rate to 0.96 and our loss function quickly decreased after training 500 K interactions. Moreover, the size of our model is 278 M and the number of our model parameters is 484,292.

## 4. Experimental Results

### 4.1. Evaluation Criteria and Comparison

Quantitative evaluation of image fusion is a non-trivial task due to the unavailability of the reference images in most cases. Hence, multiple evaluation metrics have been introduced for evaluating image fusion performance. However, there is hardly any consensus on which metrics can completely describe the fusion performance. Liu et al. [51] classified most of the existing image fusion evaluation metrics into four groups (categories) of metrics. These metrics are based on ‘image-structural similarity’, ‘image features’, ‘information theory’ and ‘human perception’. Apart from these four classifications, a new metric based on ‘visual information fidelity’ was introduced by Han et al. [52]. For a comprehensive evaluation of our proposed *MFNet* we report results on all five metrics. A description of these metrics follows:

Let *A* and *B* represent the source multi-focus images pair and let the fused image be represented by *F*.

(1) *Cvejie’s Metric $Q_{C}$* [53]. This metric mainly concerns the structural information in the source images and measures the amount of this information retained in the resulting fused image. It belongs to the ‘image-structural similarity’ category. It is defined as follows:(4)$$Q_{C}=\sum_{\omega \in W}sim\left(A,B,F|\omega \right)Q\left(A,F|\omega \right)+\left(1-sim\left(A,B,F|\omega \right)\right)Q\left(B,F|\omega \right)$$
where ω is the analysis window and *W* is the family of all the windows, *Q* is the Universal Image Quality Index, sim represents the weighting factor and depends on the similarity in the spatial domain between the input images and the fused image defined in [53].

(2) *Multi-scale Scheme Metric $Q_{M}$* [54]. Based on ‘image features’ thistmetric measures thetamount of edge information retained in the fused image from the source images. It is defined as follows:(5)$$Q_{M}=\prod_{s=1}^{S}{\left(Q_{s}^{AB/F}\right)}^{\beta_{s}}$$
where $Q_{s}^{AB/F}$ is the normalized global edge preservation value at scale *s*, *S* is the number of the decomposition scale, βs is a scale adjusting parameter [54].

(3) *Tsallis Entropy Metric $Q_{TE}$* [55]. Inspired by ‘information theory’ this metric measures the total amount of information retained by the final fused image from the source images. It is defined as follows:(6)$$Q_{TE}=\frac{I^{q}\left(A,F\right)+I^{q}\left(B,F\right)}{H^{q}\left(A\right)+H^{q}\left(B\right)-H^{q}\left(A,B\right)}$$
where $I^{q}\left(A,F\right)+I^{q}$ and $I^{q}\left(B,F\right)+I^{q}$ are the Tsallis entropies, $H^{q}\left(A\right)$ and $H^{q}\left(B\right)$ are the marginal entropies of images *A* and *B*, and $H^{q}\left(A,B\right)$ is the Tsallis Mutual Information between image *A* and *B*, *q* is a real value, and the detailed description of $Q_{TE}$ is given in [55].

(4) *Chen–Varshney Metric $Q_{CV}$* [56]. This metric is inspired by ‘human perception’ and compares the ‘edge information’ between source and fused images. It is calculated as follows:(7)$$Q_{CV}=\frac{\begin{array}{c} \sum_{l=1}^{L}\left(\lambda \left(I_{A}^{W_{l}}\right)D\left(I_{A}^{W_{l}},I_{F}^{W_{l}}\right)+\lambda \left(I_{B}^{W_{l}}\right)D\left(I_{B}^{W_{l}},I_{F}^{W_{l}}\right)\right) \end{array}}{\begin{array}{c} \sum_{l=1}^{L}\left(\lambda \left(I_{A}^{W_{l}}\right)+\lambda \left(I_{B}^{W_{l}}\right)\right) \end{array}}$$
where $D(I_{A}^{W_{l}})$ and $D(I_{B}^{W_{l}})$ is the similarity measure in the local region *W*, λ(.) represents the local region saliency value, $W_{l}$ is a local region, *L* denotes the number of nonoverlapping regions, the detailed description can be found in [56].

(5) *Visual Information Fidelity Metric VIFF* [52]. It is a ‘multi-resolution image fusion’ metric and measures the visual information fidelity transferred from input images *A* and *B* to the fused images *F*. It is defined as follows:(8)$$VIFF\left(A,B,F\right)=\sum_{k=1}^{K}p_{k}VIFF_{k}\left(A,B,F\right)$$
where *k* and *K* denote the sub-band index and the number of sub-bands, respectively. The detailed description of *VIFF* is given in [52].

Note that, for all the metrics except for $Q_{CV}$, a large value means an improved fusion quality of a fused image. In contrast, a smaller value of $Q_{CV}$ indicates a more improved result.

As mentioned earlier, *MFNet* is an *unsupervised* multi-focus image fusion technique. Hence, we can only compare the performance of our method with the available state-of-the-art *supervised* methods. These include the Non-Subsampled Contournet Transform (NSCT) [29], Guided Filtering (GF) [35], Dense SIFT (DSIFT) [33], as well as the methods based on Boundary Finding (BF) [57], Convolutional Neural Network (CNN) [18], the U-net [41]; deep unsupervised algorithms FusionDN [43], MFF-GAN [44] and U2Fusion [42]. We implemented these algorithms using code acquired from their respective authors. We carry out extensive experiments on nine pairs of multi-focus images from two public benchmark datasets: the multi-focus image fusion dataset [58] and the recently released ‘Lytro’ [48] dataset.

### 4.2. Qualitative Results

Figure 3 compares the results of our proposed *MFNet* with the other best performing multi-focus image fusion approaches on the “Clock” image set. It is evident that our proposed algorithm provides the optimum fusion result among these methods. To aid comparison, we depict the magnified regions of the fused images taken from Figure 3, as shown in Figure 4. The results clearly show that the fused images from *MFNet* contain no obvious artifact in these regions, while the fused results from the other methods contain some artifacts around the boundary of focused and defocused clocks (highlighted with green rectangles) and pseudo-edges (highlighted with red rectangles).

Detailed and magnified results of the “Volleyball Court” image set are depicted in Figure 5 and Figure 6, respectively. The fused result obtained with the BF method is distinctly blurred. Note that the fused result from the NSCT method contains artifacts (highlighted as pink rectangles) while the results of GF, DSIFT, BF, CNN and U-net algorithms betray blur artifacts around the fence edges (highlighted as green rectangles). The output of the U-net algorithm shows stark edge artifacts, especially (seen in Figure 6(h2)). Our proposed network displays no such shortcomings.

Figure 7 and Figure 8 present the original and magnified visual comparison of image fusion algorithms on the “Model Girl” image set. Although all the algorithms show similar results for the background focused region (first row of Figure 8), we can easily find blur artifacts on the girl’s shoulder in the results of NSCT, GF, DSIFT, BF, CNN and U-net algorithms. By contrast, the fused results from our method cannot find the obvious blur artifacts.

### 4.3. Quantitative Comparison

To further demonstrate the effectiveness of our proposed fusion method, we perform experiments on nine pairs of multi-focus images, as shown in Figure 9. Among them two pairs are from the grayscale Multi-focus Image dataset [18] (seen in the first two rows of Figure 9) while the remaining are from the Lytro dataset [48]. The mean and standard deviation of scores achieved by our proposed *MFNet* are reported in Table 1. The table also compares the results of our *unsupervised* MFNet with six state-of-the-art *supervised* image-fusion methods. Our proposed method outperforms the compared state-of-the-art on all metrics except the $Q_{TE}$ and $Q_{M}$ metrics where we rank second. Moreover, the $Q_{M}$ metric, our proposed method is very close to the best method NSCT. As shown in Table 1, *MFNet* achieves the optimal result on $Q_{CV}$ and the suboptimal result on $Q_{M}$ that indicates more edge information in our result inherited from the source images. Similarly, the optimal result on $Q_{C}$ and the suboptimal result on $Q_{TE}$ show that the results obtained by *MFNet* contain higher similarity with the source images with less distortion or artifacts, and more information, respectively. The results are consistent with the qualitative results shown in Figure 10. Moreover, the best result on VIFF illustrates that *MFNet* achieves the best information fidelity. What is more, we also compare our proposed *MFNet* with recently presented deep *unsupervised* methods FusionDN, MFF-GAN and U2Fusion. Table 2 gives the mean and standard deviation of the scores achieved by different *unsupervised* methods. From Table 2, it can be found that *MFNet* achieves the optimal results on $Q_{C}$, $Q_{M}$, $Q_{TE}$ and $Q_{CV}$. The optimal mean values of our proposed method on these metrics show that our results contain more edge information, more significant information, higher similarity with the source image. Furthermore, the result on VIFF shows that our method should improve the information fidelity compared with other deep unsupervised methods. Figure 11 shows the results of different *unsupervised* methods that are consistent with the qualitative results.

### 4.4. Execution Time

We used a desktop machine with 3.4 GHz Intel i7 CPU (32 RAM) and NVIDIA Titan Xp GPU (12 GB Memory) to evaluate our algorithm. For fairness, we compared our proposed algorithm with other deep unsupervised algorithms [42,43,44] since others did not perform with a GPU. The average run-time of our proposed *MFNet*, FusionDN, MFF-GAN and U2Fusion is 4.33, 1.55, 1.60 and 1.78 s, respectively. It can be found that the run-time of our proposed algorithm is higher compared to other deep unsupervised algorithms. It means that we should optimize the network to improve efficiency in future works.

## 5. Conclusions

We have introduced a new approach to learn multi-focus image fusion from an input pair of varied focus images. We have presented a deep unsupervised network (*MFNet*) that directly operates on multi-focus pair data, employs the image structural similarity (SSIM) quality metric as a loss function, and leverages the standard deviation of a local window of the image to automatically estimate the importance of the source images in the final fused image when designing the loss function. Our proposed unsupervised method obtains results that outperform, or are comparable to existing state-of-the-art supervised methods. However, our proposed method cannot directly be used to process more than two input images for the designed model and is slower than other deep unsupervised approaches. Thus, we would like to explore how to use our techniques for more than two multi-focus image fusions, and optimize our network to improve efficiency. 

## Figures and Tables

**Figure 1 sensors-20-06647-f001:**
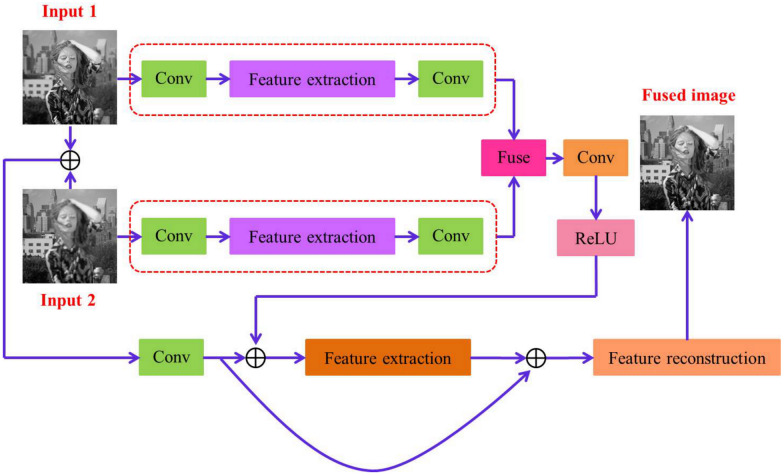
Detailed network architecture of our proposed multi-focus image fusion network. Our model constitutes three feature extraction networks for extracting non-linear features, a feature reconstruction layer for predicting the fused image, a convolutional layer for feature maps and fused features, and a transposed convolutional layer for obtaining the same dimensionality as the input image.

**Figure 2 sensors-20-06647-f002:**
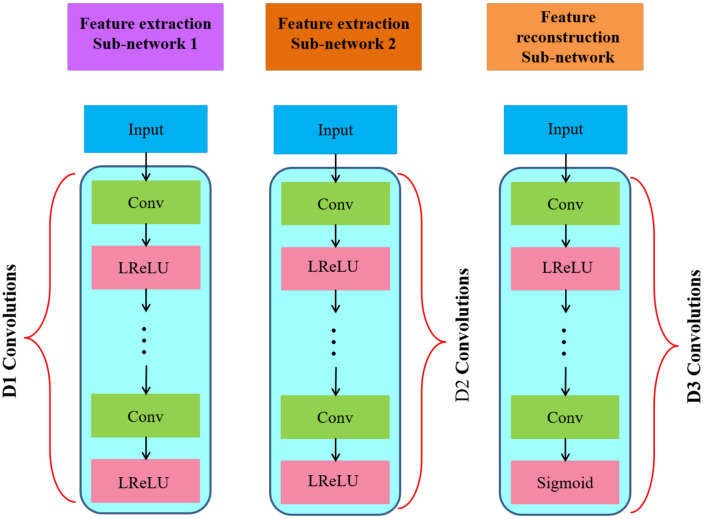
Structures of our feature extraction and reconstruction sub-networks. There are D1, D2, D3 convolutional layers in the three networks, respectively. The weights of convolutional layers are distinct among these three networks.

**Figure 3 sensors-20-06647-f003:**
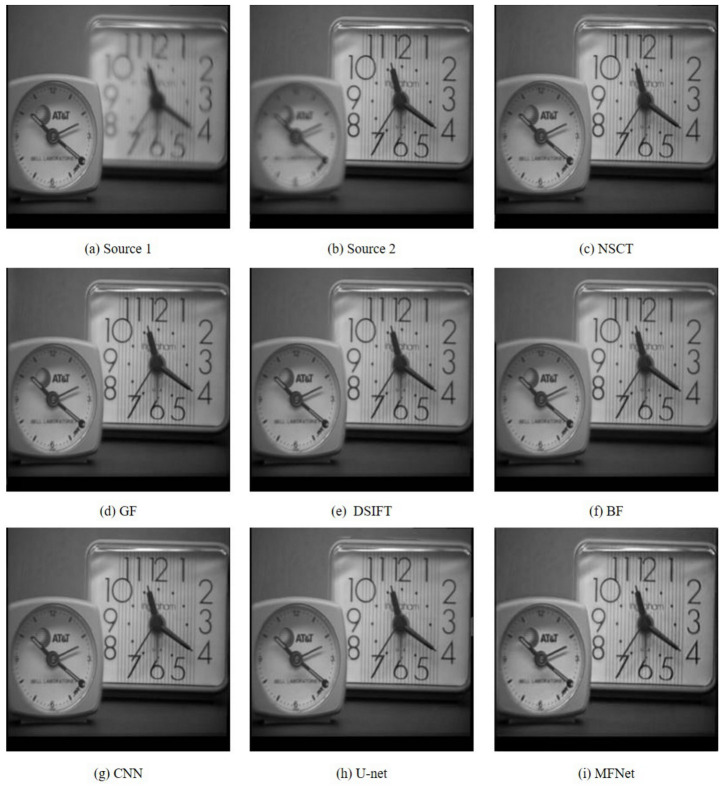
The “Clock” source image pair and their fused images obtained with different fusion methods.

**Figure 4 sensors-20-06647-f004:**
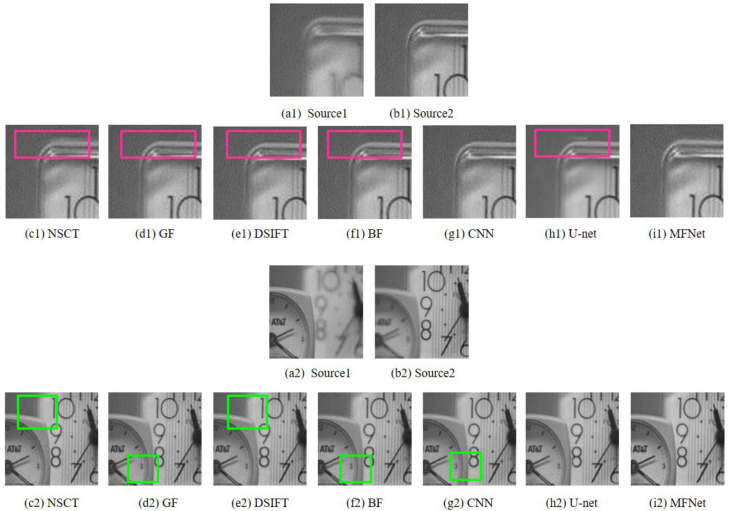
Magnified regions of the “Clock” source images and fused images obtained with different methods.

**Figure 5 sensors-20-06647-f005:**
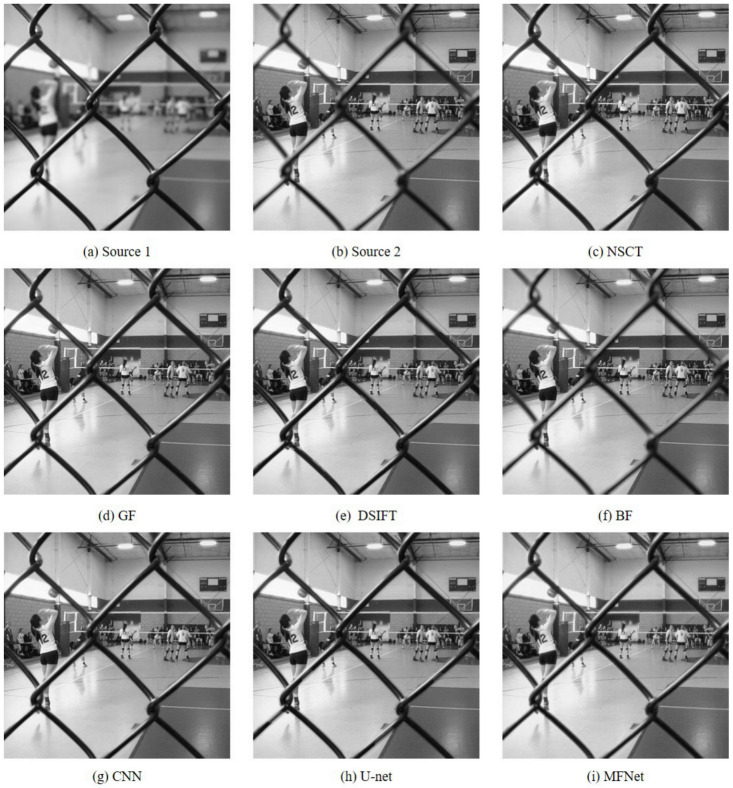
The “Volleyball Court” source image pair and their fused images obtained with different fusion methods.

**Figure 6 sensors-20-06647-f006:**
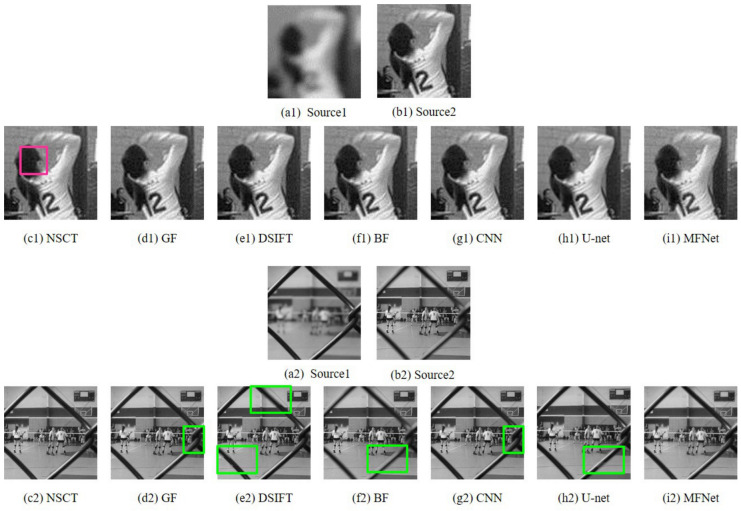
Magnified regions of the “Volleyball Court” source images and fused images obtained with different methods.

**Figure 7 sensors-20-06647-f007:**
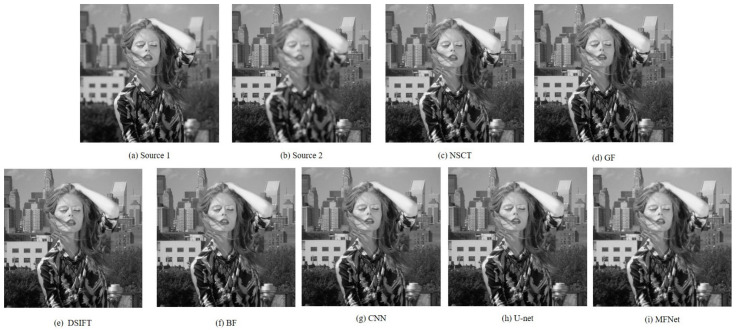
The “Model Girl” source image pair and their fused images obtained with different fusion methods.

**Figure 8 sensors-20-06647-f008:**
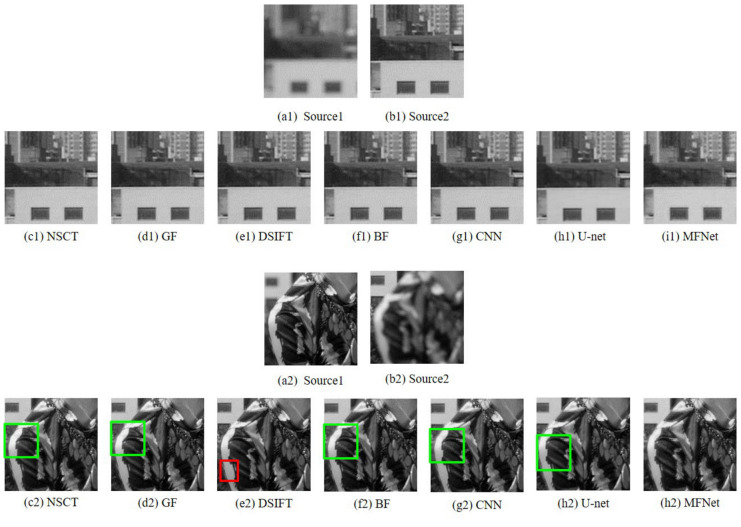
Magnified regions of the “Model Girl” source images and fused images obtained with different methods.

**Figure 9 sensors-20-06647-f009:**
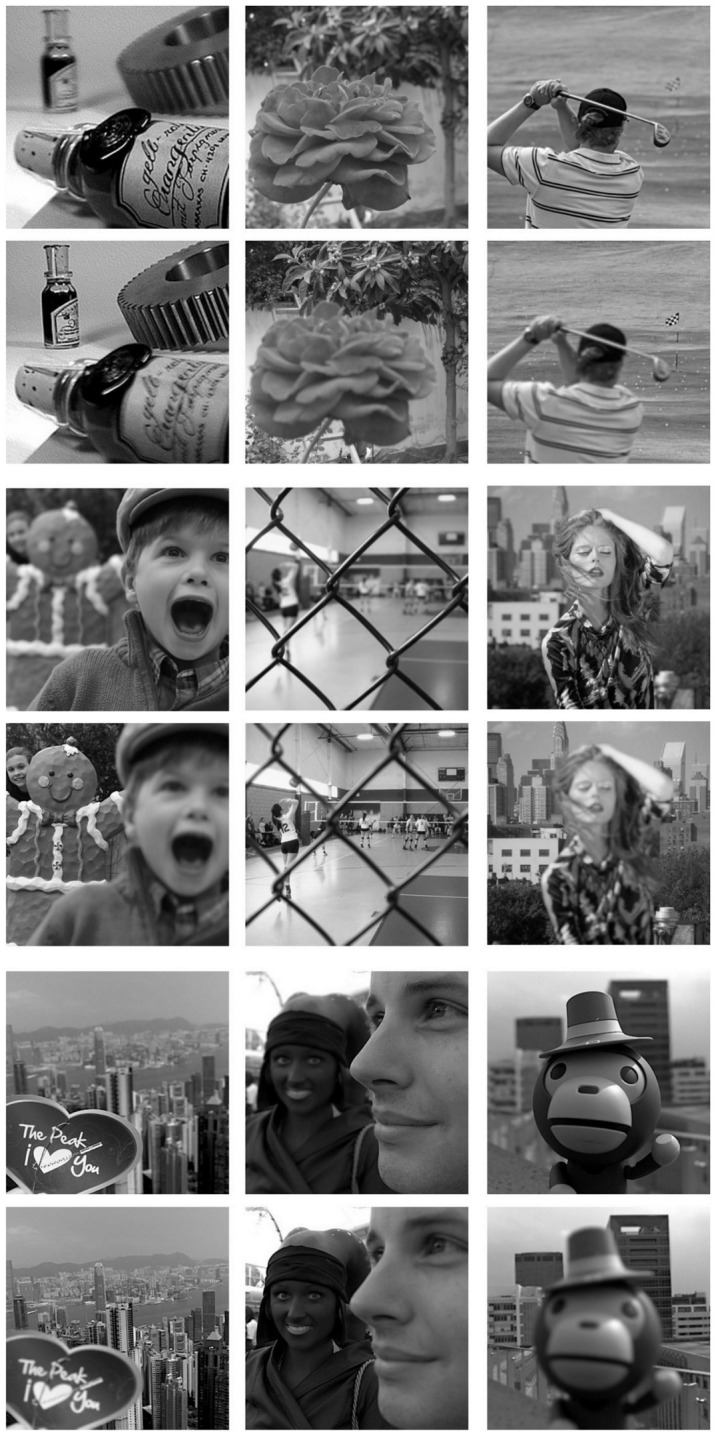
Nine pairs of multi-focus images used for validation.

**Figure 10 sensors-20-06647-f010:**
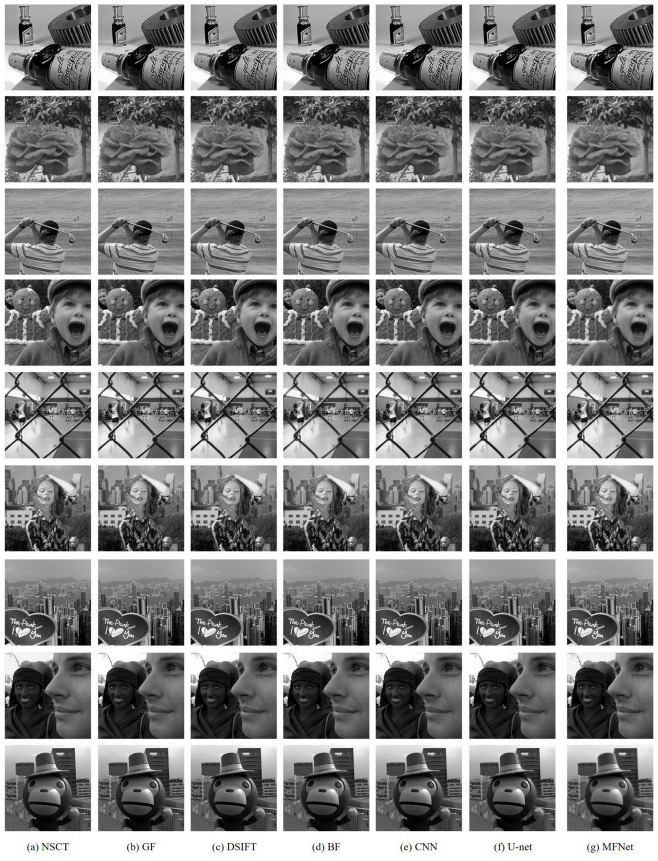
Fused results of nine pairs of source images obtained by different fusion methods.

**Figure 11 sensors-20-06647-f011:**
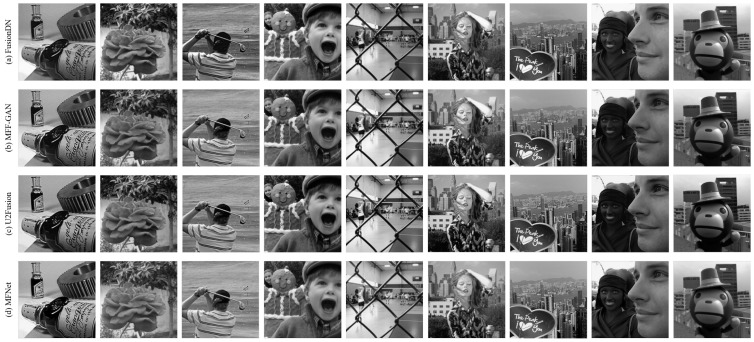
Fused results of nine pairs of source images obtained by different deep unsupervised learning fusion methods.

**Table 1 sensors-20-06647-t001:** Comparison of the mean and standard deviation (±SD) of five evaluation metrics between the *unsupervised* MFNet and the state-of-the-art *supervised* networks.

	NSCT	GF	DSIFT	BF	CNN	U-Net	MFNet
**$Q_{C}$**↑	0.788 ± 0.028	0.791 ± 0.030	0.79 ± 0.030	0.787 ± 0.033	0.794 ± 0.028	0.788 ± 0.031	**0.799** ± 0.023
**$Q_{M}$**↑	**0.934** ± 0.027	0.93 ± 0.033	0.926 ± 0.033	0.924 ± 0.033	0.929 ± 0.032	0.928 ± 0.033	0.933 ± 0.031
**$Q_{TE}$**↑	**0.929** ± 0.012	0.926 ± 0.014	0.923 ± 0.015	0.911 ± 0.026	0.925 ± 0.015	0.921 ± 0.015	0.926 ± 0.015
**VIFF**↑	0.923 ± 0.039	0.917 ± 0.041	0.918 ± 0.043	0.893 ± 0.057	0.916 ± 0.042	0.913 ± 0.040	**0.930** ± 0.060
**$Q_{CV}$**↓	54.8 ± 24.7	60.0 ± 27.3	64.0 ± 30.3	66.8 ± 34.2	61.0 ± 27.7	64.5 ± 30.0	**53.2** ± 19.9

**Table 2 sensors-20-06647-t002:** Comparison of the mean and standard deviation (±SD) of five evaluation metrics between the *unsupervised* MFNet and the state-of-the-art *unsupervised* networks.

	FusionDN	MFF-GAN	U2Fusion	MFNet
**$Q_{C}$**↑	0.7612 ± 0.027	0.7855 ± 0.020	0.7504 ± 0.028	**0.799** ± 0.023
**$Q_{M}$**↑	0.8967 ± 0.039	0.9166 ± 0.030	0.8989 ± 0.032	**0.933** ± 0.031
**$Q_{TE}$**↑	0.8930 ± 0.028	0.9176 ± 0.013	0.9113 ± 0.020	**0.926** ± 0.015
**VIFF**↑	0.9747 ± 0.129	0.9403 ±0.068	**0.9871** ± 0.071	0.930 ± 0.060
**$Q_{CV}$**↓	86.5 ± 50.8	56.6 ± 37.4	60.3 ± 44.7	**53.2** ± 19.9

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
