# Peer review of "Structural Similarity Loss for Learning to Fuse Multi-Focus Images"

_sensors, 2020, doi:10.3390/s20226647_

Round 1

Reviewer 1 Report

-The authors propose an approach to classify pixels in images as focused by using Convolutional Neural Networks. The proposal seems novel and exciting; however, some issues must be solved to give it more impact.

-In the explanation of Figure 2, the authors say that they use LReLu, but they do not reflect this matter in the Figure. (line 171)

-For clarity, it would be better if Equation 2 is rewritten in the 'If...  Otherwise' format. It could also be more elegant if the authors use some other letter (say S) instead of 'Score' in this and related equations. The same situation applies to Equation 3. (line 197)

-Is it necessary the absolute value of N in Equation 3? (line 199)

-Is there a criterion to utilize 5, 6, and 7 convolutional layers in the networks? (line 209)

-The 'Lytro' dataset has two different references.

-The authors say that 'Our proposed method outperforms the state-of-the-art on all metrics except the QTE metric where we rank second'; however, such claim does not seem to apply to measures Q_TE and Q_M. Moreover, depending on the number of runs, the direct comparison could be biased, and different statistical methods would use, such as the Wilcoxon or the Friedman tests.

Reviewer 2 Report

This paper proposes a new approach to learn multi-focus image fusion, from an input pair of varied focus images. The topic is interesting and matches well for MDPI Sensors journal. The paper contains a review of related works and shows good simulation studies.

However, the paper has some unclear points and the following minor concerns.

The authors formulate the main contribution of the paper as follows:

L48 we present an end-to-end deep network trained on benchmark multi-focus images. The proposed network takes a pair of multi-focus images and outputs the all-focus image. We train our network in an unsupervised fashion, without the need for ground truth ’all focused’ images. This method of training requires a robust loss function. We approach this problem by proposing a multi-focus Structural Similarity (SSIM) quality metric as our loss function, and use the standard deviation of a local window of the image to automatically estimate the importance of the source images in the final fused image when design the loss function.

- Thus, the authors declare that the multi-focus Structural Similarity (SSIM) quality metric as a loss function help solve the problem of the loss function robustness. However, the paper lacks simulations that confirm that exactly this metric solves this problem. An additional ablation study is needed to confirm this. The results for any other "standard" quality metric can be compared with the results for the SSIM metric.

- The paper is missing details of the Feature Extraction Sub-network architecture. In particular, the number of layers. Also, the paper would benefit if the authors made comparisons with other variants of this architecture (for example, a different number of layers) to confirm the "optimality" of their chosen architecture.

- It is necessary to provide more details in the description of the training process of the neural network. In particular, the ratio of the size of the training and test data sets and how quickly the loss decreased during training.

- It would also be useful to give the size of the resulting neural network model - the number of parameters, the file size.

L262 “The fused results from our method look aesthetically more pleasing.” - In my opinion, this phrase is too subjective. I propose to reformulate it.

L23 “Multi-Focus Image Fusion (MFIF) aims at reconstructing an 4??? "in-focus" image”

L157 “of multiple convolutional and rectification layers sans??? any pooling layer.”

Reviewer 3 Report

This paper addresses the multi-focus image fusion problem. It introduces an end-to-end machine learning approach, based on convolutional neural networks, to learn how to fuse images. The proposed formulation relies on an unsupervised learning scheme that exploits a loss function designed to encode the structural similarity of input image regions.

This paper addresses a relevant research topic, which is within the scope of the journal. The proposed solution is sound, and its description is, in general, easy to follow (specific comments related to presentation issues are provided below). The main novelty of the work relies on the proposal of a loss function which exploits structural similarity. This idea can be of interest of a large audience, including those interested in problems different from the one addressed in the paper.

Validation relies on the use of benchmark datasets and comparisons with several baselines. Both qualitative and quantitative results are reported. Observed results for the proposed method are very encouraging.

The following issues have been identified:

  1. Proofreading is needed. Some sentences are not clear enough (e.g., “As is well-known”). Grammar needs to be double-checked as well. Some examples include “…aims at reconstructing an 4 ‘infocus’…”; the use of composed words, among others.
  2. Motivational aspects presented in the introduction are not appealing. Authors are encouraged to highlight more clearly what makes the proposed solution a suitable one for the problem.
  3. The target problem is not defined clearly in the paper. Authors are encouraged to include examples to illustrate the problem. A formal definition of the problem is missing as well.
  4. It is not clear what authors mean by “D_i” in Figure 2.
  5. The evaluation protocol should consider the evaluation of other structural similarity functions. Have authors tested different window sizes?
  6. Evaluation criteria are not presented clearly. Several components of Equations 4-8 are not defined.
  7. As stated in the related work section, references [42-44] introduced unsupervised approaches for the same problem. The proposed approach needs to be compared with those approaches. It is not clear as well the pros and cons of those formulations when compared with the proposed approach.
  8. The discussion of quantitative results needs to be expanded. What can we learn from the results observed for the different assessment criteria used?
  9. The paper lacks discussion of cases of failure and limitations of the proposed approach.
  10. It is not clear why the execution time of the proposed approach is contrasted only with the one computed for the method described in [18].

Round 2

Reviewer 2 Report

I believe that this updated version of the paper can be published in the journal.

Reviewer 3 Report

The authors have addressed most of the raised issues properly.